# Higher-Order Cognitive Chain-of-Thought Is Enough: Evaluation, Analysis, and Optimization

## Abstract

Chain-of-Thought (CoT) data has become essential for advancing large language models' reasoning capabilities, yet current quality assessment methods neglect the quality of underlying reasoning processes and thus undermine their effectiveness. To address these challenges, we propose a CoT data quality assessment framework from a cognitive perspective, grounded in Bloom's Taxonomy as our core theoretical foundation. Through systematic analysis of existing CoT datasets, we reveal that current CoT data exhibits significant distributional biases toward intermediate-order cognitive operations, failing to adequately represent the full spectrum of human-level cognitive capabilities. These findings demonstrate systematic inadequacies in reasoning quality across multiple benchmarks, with models struggling to reproduce sophisticated cognitive processes essential for complex problem-solving. Based on these insights, we propose a simple-yet-effective cognitive-guided CoT data enhancement approach that supplements datasets with minimal higher-order cognitive CoT data. Consequently, we introduce a simple-yet-effective CoT data enhancement method that rapidly enhances model performance using minimal additional high-order cognitive CoT data, experiments demonstrates the effectiveness of cognitive-aware CoT dataset construction and evaluation.

## 1 Introduction

Chain-of-Thought (CoT) data plays a pivotal role in advancing large language models (LLMs). Besides CoT data significantly enhances the accuracy of complex problem-solving by providing structured reasoning examples (Wei et al., 2022; Zhang et al., 2023; Wei et al., 2022; Suzgun et al., 2023), it effectively induces models to develop sophisticated reasoning capabilities, especially in reinforcement learning (RL) based models like Deepseek-R1 (DeepSeek-AI et al., 2025). By incorporating CoT data, RL-based reasoning LLMs can learn to break down complex problems into manageable steps (DeepSeek-AI et al., 2025; Wen et al., 2025), which not only boosts performance but also enhances the interpretability of the model's decision-making process, making CoT data invaluable for improving the overall effectiveness and trustworthiness of these advanced LLMs. Hence, constructing and employing CoT datasets becomes an important task (Zhang et al., 2023; Suzgun et al., 2023; Yin et al., 2025).

Researchers have developed various CoT data construction methodologies, primarily encompassing two technical approaches: direct utilization of pre-existing CoT datasets and acquisition via knowledge distillation (Wang et al., 2023a; Chen et al., 2023). Direct utilization typically involves manual annotation, wherein domain experts meticulously construct high-quality reasoning chains (Li et al., 2023a). Conversely, knowledge distillation leverages large pre-trained language models to automatically generate CoT data. To ensure CoT data quality, these methods introduce additional verification mechanisms to filter out low-quality CoT data, such as teacher-generated rationales with post-hoc filtering (Li et al., 2023b), self-distillation via self-consistency that retains rationales yielding consistent answers (Wang et al., 2023b), and verification-based distillation that employs explicit verifiers to validate intermediate reasoning steps (Ling et al., 2023).

However, despite the proliferation of CoT construction efforts, we argue that current CoT datasets suffer from fundamental quality assessment challenges that undermine their effectiveness. Current research exhibits a pronounced trend toward scalability, with newly constructed CoT datasets not only growing in data volume but also in length (Wang et al., 2023b; Fang et al., 2025; Golovneva et al., 2023; Nguyen et al., 2024). This "bigger and longer" development strategy lacks robust quality assessment foundations, implying three fundamental limitations. **(1) Existing evaluations ignore the quality of reasoning processes.** Existing metrics concentrating exclusively on length and accuracy cannot adequately characterize the semantic and cognitive properties intrinsic to reasoning chains; they induce fundamental gaps in assessing the cognitive essence of the reasoning processes Prasad et al. (2023). **(2) Existing evaluations may introduce systematic biases.** Longer CoT frequently exhibits extraneous content, erroneous logical steps, or task-irrelevant information, whereas cognitively sound reasoning is characteristically concise and demonstrates direct problem-solving efficacy (Ling et al., 2023; Li et al., 2023a; Wang et al., 2023b). **(3) Existing evaluation metrics fail to reflect data quality defects in terms of model performance.** Emergent empirical evidence indicates that neither length nor scale of CoT data correlates with substantive reasoning capability gains Kejriwal et al. (2024); Sylolypavan et al. (2023).

To address these limitations, cognitive theories can be leveraged for CoT quality assessment. First, cognitive theory inherently prioritizes the quality of reasoning processes. Since CoT data fundamentally aligns with human cognitive operations, cognitive theory provides normative frameworks for assessing the quality of intermediate reasoning operations (Wei et al., 2022; Nguyen et al., 2024). Second, cognitive theory emphasizes the depth of processing, establishing normative quality benchmarks that mitigate quality degradation resulting from superficial pursuit of length or scale metrics, thereby ensuring evaluation criteria correspond to authentic cognitive rigor. Finally, cognitive science has established robust causal links between hierarchical cognitive operations and performance efficacy, providing mechanistic foundations for fine-grained diagnostic evaluations of model reasoning capabilities.

Specifically, we employ Bloom's Taxonomy Bloom et al. (1956) as our core cognitive framework, a theory that categorizes cognitive abilities into six cognitive levels (remembering, understanding, applying, analyzing, evaluating, and creating), which is particularly suited for systematically parsing the types of cognitive operations within CoT. We have developed methods to automatically map CoT steps to Bloom's Taxonomy levels and constructed an innovative evaluation metric system based on this mapping, focusing on four key research questions:

**RQ1:** How well do current CoT datasets represent compared to human cognitive processes?

**RQ2:** Can LLMs reproduce human-level cognitive processes?

**RQ3:** What are the characteristics of the cognitive processes exhibited by LLMs?

**RQ4:** How can CoT datasets be optimized to enhance their cognitive expressiveness?

The main contributions of this work are summarized as follows:

- To the best of our knowledge, we are the first to systematically investigate CoT data quality assessment from a cognitive perspective, establishing a novel evaluation paradigm grounded in cognitive principles.

- We develop a comprehensive evaluation framework anchored in Bloom's Taxonomy that enables precise quantification of cognitive complexity within CoT reasoning sequences, providing both accuracy and computational efficiency.

- We conduct extensive experiments to analyze the quality of existing CoT datasets and knowledge-distilled CoT data, revealing systematic biases and cognitive inadequacies that compromise reasoning quality across multiple benchmarks.

- We propose a simple-yet-effective cognitive-guided approach for CoT data enhancement, with experimental validation demonstrating substantial improvements in model reasoning capabilities using little CoT data.

## 2 RELATED WORK

**CoT Datasets**   Recent efforts in CoT dataset construction have emphasized creating coherent reasoning chains across diverse tasks and domains, thereby enhancing the interpretability and logical capabilities of LLMs. Early large-scale efforts, such as ThoughtSource(Ott et al., 2023), aggregate CoT examples from multiple domains, including science, mathematics, medicine, and general question answering, while standardizing formats and providing tools for evaluation, fine-tuning, and reasoning analysis. Other works such as LogiCoT(Liu et al., 2023) and OmniThought(Cai et al., 2025), introduce large-scale CoT dataset and show the effectiveness of them. Mathematics has emerged as a central domain for CoT research, with datasets(Yu et al., 2024; Ling et al., 2017; Amini et al., 2019; Patel et al., 2024; Mitra et al., 2024) providing complementary resources with varying scale, annotation format, and reasoning granularity, supporting tasks such as symbolic manipulation, algebraic reasoning, and multi-hop deduction., and reverse reasoning to enhance existing benchmarks (Cobbe et al., 2021; Lightman et al., 2024). Generally, CoT datasets have progressed toward broader domain coverage and more sophisticated reasoning processes.

**CoT Data Evaluation**   Recent research has increasingly focused on assessing the quality of CoT data. Numerous studies have attempted to quantify and evaluate the complexity of CoT data through various approaches. For instance, Budagam et al. (2024) analyzed CoT data complexity using statistical indicators such as length, entropy, and logical leaps, while Li et al. (2024) introduced structural entropy and cognitive tree representations. Alternative methods have examined reasoning complexity through semantic stability (Yao et al., 2023) and logical consistency (Xu et al., 2025), employing embedding trajectory geometry and logical template matching as key analytical tools. Additionally, Wang et al. (2024) framed model generation as progressive cognitive unfolding, introducing the Local Disbalance Rate metric to capture the interplay between cognitive complexity and generative behavior. Notably, research has revealed that increasing reasoning chain length or depth does not guarantee improved performance (Lee et al., 2025; Kang et al., 2024). Overall, existing CoT data evaluation remains predominantly focused on analyzing surface-level complexity of reasoning chains, lacking deeper insights into their underlying cognitive capabilities.

## 3 METHODOLOGY

### 3.1 BLOOM'S TAXONOMY FRAMEWORK

Bloom's Taxonomy serves as our theoretical framework for analyzing the quality of CoT datasets. Bloom's Taxonomy, first introduced by Bloom et al. (1956) and Anderson & Krathwohl (2001), provides a hierarchical classification of cognitive processes encompassing six levels: Remembering, Understanding, Applying, Analyzing, Evaluating, and Creating, alongside four knowledge dimensions: Factual, Conceptual, Procedural, and Metacognitive.

We selected Bloom's Taxonomy as our theoretical foundation for three reasons. (1) Bloom's Taxonomy describes cognitive operational processes, which perfectly align with the core requirement of CoT data assessment, to reveal the connection between reasoning processes and quality. (2) Bloom's Taxonomy provides a structured framework with dual-dimensional structure (cognitive processes and knowledge types), enabling precise characterization of the cognitive-knowledge coupling patterns, thereby offering an interpretable mechanism for model performance. (3) Bloom's Taxonomy possesses a comprehensive conceptual system and standardized assessment dimensions that ensure both theoretical scientific reliability and operational standards convertible to automated evaluation.

### 3.2 PROBLEM DEFINITION

To systematically capture the cognitive structure of reasoning in CoT data, we formalize a taxonomy-based annotation scheme. This scheme leverages Bloom's Taxonomy to evaluate both the inherent cognitive difficulty of the instructions and the step-wise cognitive progression of reasoning. By representing CoT annotation through a formal mathematical expression, we can explicitly define the procedure and goals of the annotation task, thereby further enabling a clear and rigorous characterization of both the task itself and the intended annotation outcomes. The set of cognitive levels and

knowledge levels in Bloom's Taxonomy are denoted as:

$$\mathcal{C} = \{\text{REMEMBERING, UNDERSTANDING, APPLYING, ANALYZING, EVALUATING, CREATING}\}$$

$$\mathcal{K} = \{\text{FACTUAL, CONCEPTUAL, PROCEDURAL, METACOGNITIVE}\}.$$

Consider giving a CoT instance $x = (\text{Q}, CoT, \text{A})$, where Q denotes the input question, $CoT = (s_1, s_2, \ldots, s_n)$ is the generated reasoning chain consisting of intermediate steps $s_i$, A is the final answer. Then, we present two cognitive annotation tasks with different granularities: **instruction and trajectory cognitive annotation**.

**Definition 3.1 (Instruction Cognitive Annotation)** *The **instruction cognitive annotation** assigns a single cognitive label $c^* \in \mathcal{C}$ and a knowledge label $k^* \in \mathcal{K}$ to a CoT instance, reflecting the highest cognitive level required to correctly respond to the given instruction. Formally, given the CoT, question A, answer A, and the annotation prompt $P_I$, the label is defined as*

$$L_I(CoT \mid Q, A, P_I) = (c^*, k^*, t_I^*), \tag{1}$$

*where $t_I^*$ represents the reasoning process used in the annotation, capturing the annotator's thought process in determining the highest cognitive level required to answer the instruction.*

For example, a simple factual recall problem may be labeled as REMEMBERING, whereas a multi-step logical deduction problem may require ANALYZING or higher.

**Definition 3.2 (Cognitive Trajectory Annotation)** *The **cognitive trajectory annotation** captures the cognitive labels of individual reasoning steps within a CoT. Each reasoning step $s_i \in CoT$ is paired with a cognitive label $c_i \in \mathcal{C}$, producing a cognitive trajectory $(c_1 \rightarrow c_2 \rightarrow \cdots \rightarrow c_n)$ that aligns one-to-one with the reasoning steps within the annotation prompt $P_T$:*

$$L_T(CoT \mid Q, A, P_T) = (\tau, t_T^*), \tag{2}$$

$$\tau = (c_1 \rightarrow c_2 \rightarrow \cdots \rightarrow c_n), \tag{3}$$

*where $t_T^*$ denotes the annotator's thought process followed during annotation, reflecting how each step $s_i$ was analyzed to determine its cognitive label $c_i$.*

The proposed annotation scheme not only formalizes cognitive labeling, but also enables structural analysis of CoT sequences from two complementary perspectives:

**Depth** The highest cognitive level reached in the reasoning chain, reflecting overall problem difficulty;

**Progression** The coherence of cognitive steps across the chain, indicating whether reasoning follows a consistent trajectory or shows regressions/redundancies.

Together, these two perspectives support a fine-grained evaluation of reasoning quality, considering both the sophistication and the structural integrity of the thought process.

### 3.3 COGNITIVE ANNOTATION WITH LLMS

We adopt *Qwen2.5-72B-Instruct* as the primary annotator for cognitive-level labeling, inspired by the strong evaluation results of *CompassJudger-32B-Instruct* (Cao et al., 2024). Specifically, we use the instruction-tuned *Qwen2.5-72B-Instruct* as a few-shot annotator for both model-generated and human-written CoTs. The annotation prompts are constructed by combining concise operational definitions of Bloom's cognitive processes and knowledge dimensions with representative examples, thereby grounding the labeling in explicit decision rules.

To assess reliability, we randomly select 1,000 CoT instances, and engage three cognitive science experts for independent annotation. We compute Cohen's Kappa ($\kappa$) against three experts, and the results are shown in Table 1. **Qwen2.5-72B-Instruct achieves the highest agreement across both knowledge and cognitive dimensions, with $\kappa > 0.88$ indicating near-human consistency and demonstrating the effectiveness of the model-based annotation pipeline.** The consistency evaluation result also demonstrates the effectiveness of model-based annotation, details are shown in Appendix 7.

Table 1: Cohen's Kappa ($\kappa$) consistency scores computed on 1,000 sampled CoT instances, comparing multiple LLM annotators with three experts.

| | Knowledge Dimension | | | | Cognitive Dimension | | | |
|---|---|---|---|---|---|---|---|---|
| | Expert 1 | Expert 2 | Expert 3 | Average | Expert 1 | Expert 2 | Expert 3 | Average |
| Qwen2.5-32B-Instruct | 0.9758 | 0.9801 | 0.9700 | 0.9753 | 0.8533 | 0.8718 | 0.8649 | 0.8633 |
| Qwen3-32B | 0.9646 | 0.9707 | 0.9609 | 0.9654 | 0.7811 | 0.7754 | 0.7811 | 0.7792 |
| CompassJudger-2-32B-Instruct | 0.9818 | 0.9790 | 0.9794 | 0.9801 | 0.8383 | 0.8517 | 0.8424 | 0.8441 |
| **Qwen2.5-72B-Instruct** | **0.9874** | **0.9874** | **0.9868** | **0.9872** | **0.8820** | **0.8832** | **0.8914** | **0.8855** |
| **ChatGPT4-mini** | 0.9646 | 0.9707 | 0.9609 | 0.9654 | 0.7811 | 0.7754 | 0.7811 | 0.7792 |

## 3.4 METRICS

For quantifying the CoT data quality of a dataset, we define $\bar{V}^c$ and $\bar{V}^k$ as the mean values for the cognitive and knowledge dimensions, respectively, and denote $\epsilon^c$ and $\epsilon^k$ as their respective standard deviations. Specifically, following Bloom's Taxonomy hierarchy from low to high, we annotate each CoT instance with cognitive dimension values from 1 to 6, where 1 denotes the lowest cognitive level (Remembering) and 6 represents the highest level (Creating). Similarly, we annotate the knowledge dimension values from 1 to 4, ranging from lower-order to higher-order knowledge types.

We further introduce a new metric **Bloom's Taxonomy Weighted Level Index** $W_c$ and $W_k$ in order to quantify the distance between each cognitive and knowledge dimension's distribution and the uniform distribution. Specifically, we define $W_c = \min(N_C \cdot p_c, 1)$, $W_k = \min(N_K \cdot p_k, 1)$, where $N_C = 4, N_K = 6$ are the number of dimensions of cognitive and knowledge, respectively, and $p_c, p_k$ are its relative frequency. The $\min$ function ensures the metric to 1 for dimensions with $p_k \leq \frac{1}{N_K}$ and $p_c \leq \frac{1}{N_C}$.

## 4 EXPERIMENT

To evaluate and improve the quality of existing CoT datasets, we conduct a series of experiments to answer the following research questions:

**RQ1:** How well do current CoT datasets represent compared to human cognitive processes?

**RQ2:** Can LLMs reproduce human-level cognitive processes?

**RQ3:** What are the characteristics of the cognitive processes exhibited by LLMs?

**RQ4:** How can CoT dataset be optimized to enhance their cognitive expressiveness?

## 4.1 EXPERIMENT SETUP

**Datasets** We evaluate the quality of 10 different widely-used CoT datasets (Cobbe et al., 2021; Mahdavi et al., 2025; Albalak et al., 2025; Lu et al., 2024; Gao et al., 2025; Hendrycks et al., 2021; Ling et al., 2017), which can be categorized into 2 groups according to their usage. **(1) Human-authored CoT datasets.** We assembled a diverse collection of human-written CoT corpora. The datasets cover a wide range of mathematical reasoning tasks, from standard exercises to Olympiad-level problems, and constitute the primary source of high-quality human CoTs that capture natural cognitive processes. **(2) LLM-distilled CoT datasets.** We collected distilled CoTs from state-of-the-art models. Distillation was performed on representative benchmarks selected for their domain coverage and ability to reflect realistic model reasoning. Details of the CoT datasets are shown in Appendix (Table 2 and Table 3).

**Settings** Our experiments are conducted on 4 accelerated GPUs, and we use PyTorch 2.6 in Python 3.11. We set the maximum sequence length for both input and output sequences to 1536 tokens.

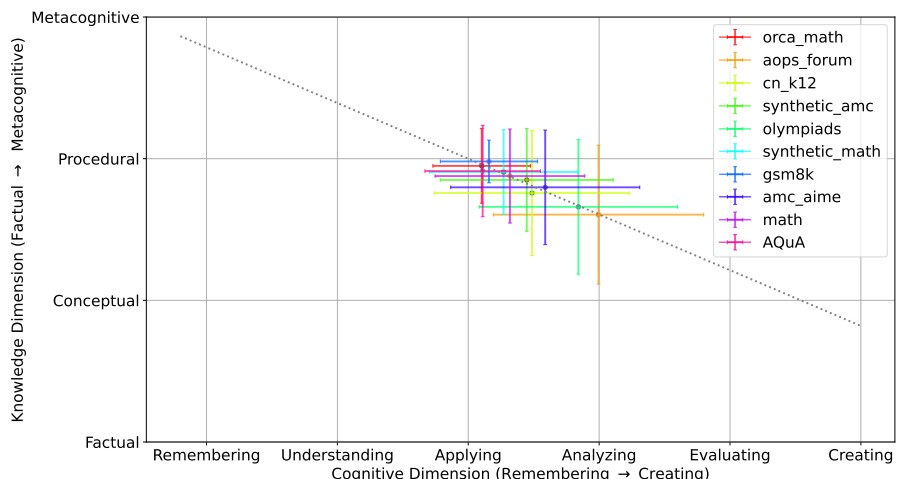

Figure 1: The distribution of current CoT data across Bloom's two dimensions.

## 4.2 ANALYSIS

### 4.2.1 RQ1: HOW WELL DO CURRENT COT DATASETS REPRESENT COMPARED TO HUMAN COGNITIVE PROCESSES?

**Observation 1: Current CoT datasets demonstrate systematic bias toward intermediate-order cognitive processes and knowledge types.** Figure 1 illustrates that most CoTs fall between Applying and Analyzing, indicating an emphasis on intermediate-order cognitive reasoning. However, current CoT datasets exhibit marked deficiency in higher-order cognitive operations (Evaluating, Creating). As shown in Figure 2a, a minimal proportion of CoT data operates at higher-order cognitive levels (Evaluating and Creating). Similarly, Figure 1 reveals a pronounced concentration of existing CoT datasets on conceptual and procedural knowledge types. Figure 2b further demonstrates that the vast majority of reasoning chains operate at the conceptual and procedual knowledge level, higher-order knowledge (metacognitive) remains markedly underrepresented across all evaluated datasets.

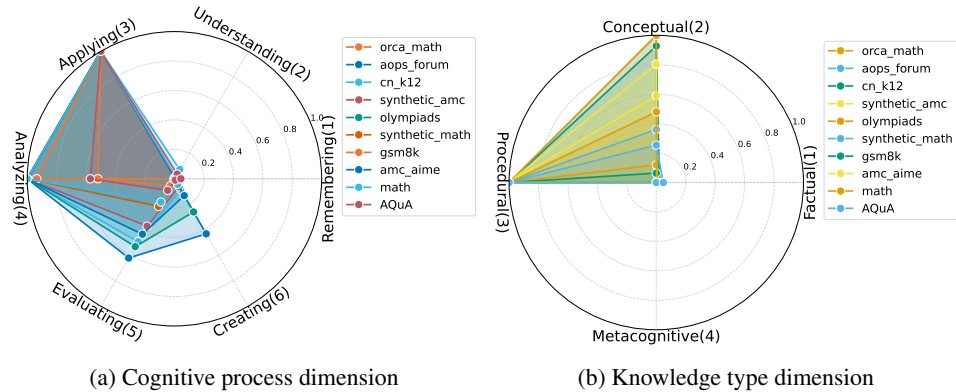

(a) Cognitive process dimension        (b) Knowledge type dimension

Figure 2: Distribution of cognitive process and knowledge types in CoT datasets.

**Observation 2: Cognitive-Knowledge dimensional misalignment.** Existing CoT datasets exhibit systematic mismatches between cognitive and knowledge dimensions that contradict human learning patterns. While optimal cognitive development follows a hierarchical progression where cognitive complexity increases with knowledge abstraction(Krathwohl, 2002), Figure 1 shows anomalous pairings, notably Applying-Procedural and Analyzing-Conceptual, which violate this fundamental cognitive-pedagogical principle.

> **Highlights 1**
>
> Current CoT datasets suffer from severe distributional imbalance and systematic inconsistency with human cognitive patterns.

### 4.2.2 RQ2: CAN LLMS REPRODUCE HUMAN-LEVEL COGNITIVE PROCESSES?

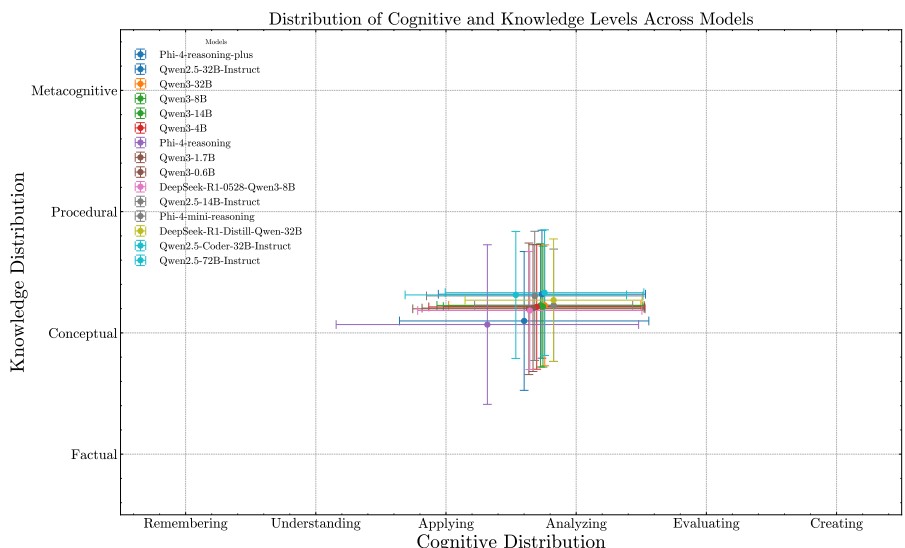

Figure 3: The distribution of distilled CoT data across Bloom's two dimensions.

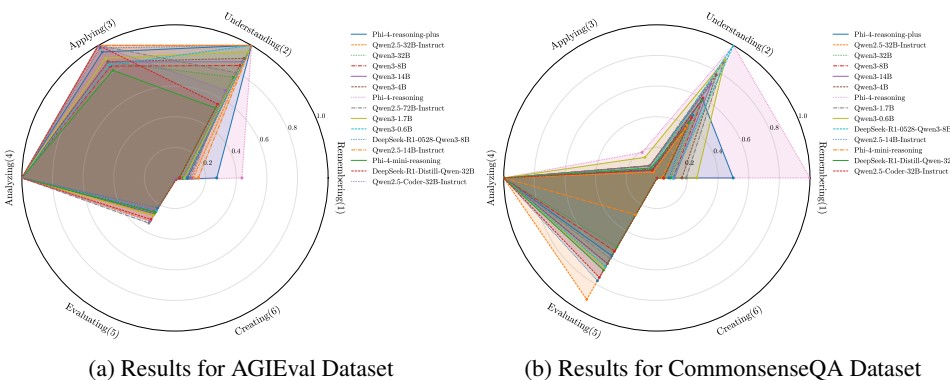

(a) Results for AGIEval Dataset                (b) Results for CommonsenseQA Dataset

Figure 4: The details for the LLM's distilled CoT results.

**Observation 3: Dimensional Concentration.** Consistent with existing CoT datasets, distilled CoT data exhibits systematic concentration in intermediate cognitive processes and knowledge dimensions, replicating the distributional bias observed in current CoT datasets. As shown in Figure 3, distilled CoTs concentrate in Applying and Analyzing operations, with sparse representation in both lower-order (Remembering, Understanding) and higher-order (Evaluating, Creating) operations. In the knowledge dimension, most models rely on Conceptual and Procedural knowledge, rather than factual recall or metacognitive reasoning.

**Observation 4: Task-Specific Diversity.** Distilled CoT data demonstrates exceptionally high diversity with pronounced distributional inconsistency across different task domains, indicating significant task-dependent variation in cognitive-knowledge patterns. We show the distribution across four different task categories in Figure 4, including professional exams, commonsense reasoning, mathematics, and code reasoning. In AGIEval and CruxEval, reasoning concentrates on Understand-

ing and Applying, with Analyzing as a secondary component. CommonsenseQA displays higher proportions of Analyzing and Evaluating, reflecting increased critical assessment demands, while Omni-MATH emphasizes Applying and Analyzing, highlighting procedural and problem-solving reasoning. Detailed experimental results are shown in Appendix (Figure 7).

**Observation 5: Human-Aligned Coupling.** Figure 3 shows that distilled CoT data achieves alignment with human cognitive patterns through proper dimensional coupling: lower-order cognitive processes consistently pair with lower-order knowledge types, while higher-order cognitive operations align with higher-order knowledge domains.

> **Highlights 2**
>
> While distilled CoT data exhibits high diversity and consistency with human cognition, its pronounced distributional imbalance critically constrains overall data quality.

### 4.2.3 RQ3: WHAT ARE THE CHARACTERISTICS OF THE COGNITIVE PROCESSES EXHIBITED BY LLMS?

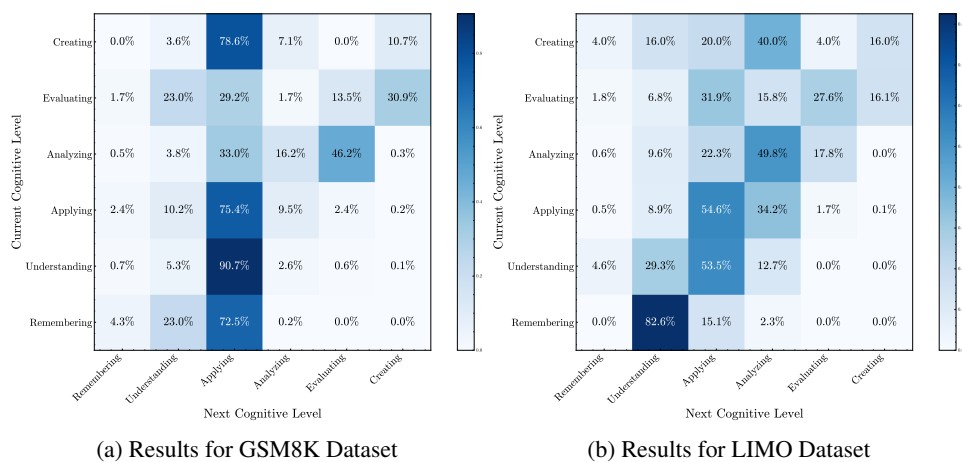

(a) Results for GSM8K Dataset  (b) Results for LIMO Dataset

Figure 5: Cognitive trajectory annotation results.

Figure 5 presents cognitive trajectory matrices for 2 different types of datasets, where each cell represents the transition probability from one cognitive level to the next in CoT reasoning, as annotated using Bloom's Taxonomy. This visualization captures the sequential cognitive dynamics of model-generated reasoning, showing how steps evolve across tasks. More detailed results are shown in Appendix (Figure 9).

**Observation 6: Cognitive Hierarchy Compliance.** All datasets demonstrate cognitive transitions that adhere to human cognitive hierarchies, with lower-order cognitive processes (Remembering/Understanding) predominantly transitioning to adjacent or higher-level cognitive processes, while reverse transitions (e.g., from Analyzing back to Remembering) occur infrequently, confirming alignment with established cognitive progression patterns (Wei et al., 2022; Huber & Niklaus, 2025).

**Observation 7: Task-Specific Transition Characteristics.** Different datasets exhibit distinct cognitive transition signatures reflecting their underlying task nature: GSM8K shows pronounced Understanding-Applying transitions (90.7%), indicative of algorithmic problem-solving characteristics; AQuA demonstrates strongest Understanding-Applying patterns (73.5%) with more distributed transitions, reflecting quantitative reasoning complexity; while LIMO exhibits strong Remembering-Understanding transitions (82.6%) coupled with notable Analyzing-Analyzing self-loops (49.8%), characteristic of iterative logical reasoning processes.

> **Highlights 3**
>
> While distilled CoT data exhibits strong task-specificity and human-aligned cognitive transition hierarchies, it inevitably suffers from persistent distributional biases.

#### 4.2.4 RQ4: How can the cognitive processes of LLMs be optimized to enhance their cognitive expressiveness?

Figure 6 illustrates the performance trends of instruction fine-tuning on three representative benchmarks, including CMMLU, MMLU-Pro (Broad Domain Knowledge), CommonsenseQA (Commonsense Reasoning), and AGIEval (Professional Exams). We compare the effect of high-order CoT instructions (corresponding to Analyzing, Evaluating, Creating in Bloom's Taxonomy) with low-order CoT instructions (corresponding to Remembering, Understanding, Applying). We present the average accuracy of all evaluations in the figure.

The results show distinct trends: for high-order cognitive level CoTs, the accuracy initially increases as the dataset size grows, but gradually saturates after around 400–600 instructions. In contrast, low-order cognitive level CoTs exhibit an increase in accuracy at small scales, but performance begins to decline as more data are added. This indicates that fine-tuning with only low-order cognitive level CoTs cannot sustainably improve LLM reasoning ability. These findings suggest that **a relatively small but high-quality subset of high-order cognitive level CoTs (approximately 300 examples) is sufficient to boost LLM performance across diverse benchmarks**, whereas simply enlarging low-order cognitive level CoT datasets may even harm performance. This highlights that a small but cognitively rich set of instructions can serve as a "golden dataset" for instruction fine-tuning, offering an efficient strategy for enhancing reasoning capabilities in LLMs.

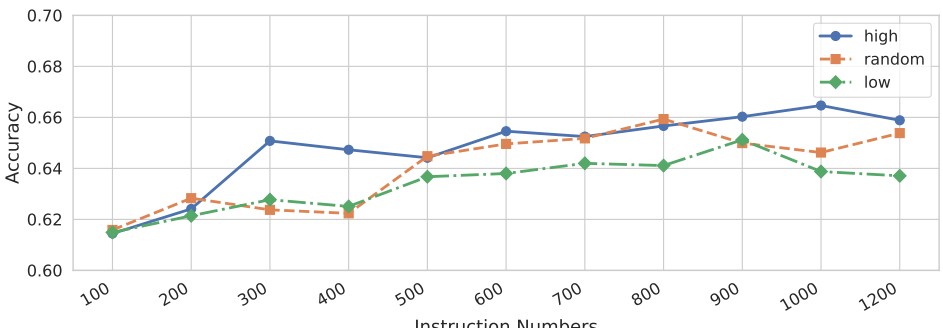

Figure 6: The fine-tuning results on Qwen2.5-7B-Base.

> **Highlights 4**
>
> Supplementing minimal amounts of high-order CoT data enables rapid model performance enhancement and stabilization.

## 5 Conclusion

In this paper, we systematically investigate CoT data quality assessment from a cognitive perspective and propose a comprehensive evaluation framework grounded in Bloom's Taxonomy. Our analysis reveals that existing CoT datasets typically suffer from distributional biases that fail to adequately represent human-level cognitive capabilities. Consequently, we introduce a simple-yet-effective CoT data enhancement method that rapidly enhances model performance using minimal additional high-order cognitive CoT data. Future efforts must prioritize the creation and integration of higher-order and metacognitive CoT data to unlock the full potential of language models for creative and reflective tasks.

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

# 6 DATA COLLECTIONS

To address different research questions, we compiled a variety of datasets from multiple domains. These datasets include human-constructed datasets, model-distilled datasets, and model fine-tuning datasets. Detailed examples of the datasets can be found in Table 2, Table 3, and Table 4.

Table 2: Human-authored Chain-of-Thought datasets.

| Dataset | Domain | Scale | Source | Usage |
|---------|--------|-------|--------|-------|
| Orac_math | Mathematics | ∼200K | Human + synthetic | Training / CoT examples |
| AOPS_forum | Math contest problems | Thousands | AoPS forum | Multi-step reasoning |
| CN_K12 | K-12 math (Chinese) | Thousands | Textbooks / exams | Cross-lingual tasks |
| Synthetic_AMC | AMC-style math | Thousands–10K | Synthetic generation | Contest-style reasoning |
| Olympiads | Math Olympiad | Thousands | Exam archives | Hard math reasoning |
| Synthetic_math | General synthetic math | >100K | Synthetic generation | Broad math reasoning |
| GSM8K | Grade-school math | 8,000 | Human-authored | Benchmark / evaluation |
| AMC_AIME | AMC + AIME contests | Thousands | Contest archives | Hard math reasoning |
| MATH | Math competition dataset | 12,500 | MATH benchmark | Step-by-step reasoning |
| AQuA | Algebra/word problems | Thousands | Human-authored | Logic + math reasoning |

Table 3: Model-distilled Chain-of-Thought datasets.

| Dataset | Domain | Scale | Usage |
|---------|--------|-------|-------|
| AQuA | Math/logic QA | Thousands | Distilled CoT training |
| GSM8K | Grade-school math | 8,000 | Distilled CoT training |
| LIMO | Logic + math reasoning | Thousands | Distilled reasoning analysis |
| MATH | Extended MATH | Thousands | Complex reasoning |

Table 4: Evaluation benchmarks for LLM performance.

| Dataset | Domain | Scale | Source | Usage |
|---------|--------|-------|--------|-------|
| AGIEval | Expert knowledge + math | Thousands | Exam-based | Cross-domain evaluation |
| MMLU-Pro | Broad expert knowledge | Thousands | MMLU series | General evaluation |
| CMMLU | Chinese / multilingual MMLU | Thousands | Exam-based | Multilingual reasoning |
| CommonSenseQA | Commonsense reasoning | 12K | CommonsenseQA benchmark | Commonsense evaluation |

# 7 ANNOTATION CONSISTENCY ANALYSIS

## 7.1 CONSISTENCY EVALUATION

Given the inherent ambiguity and subjectivity involved in annotating cognitive levels from CoT reasoning, it is essential to assess the consistency and reliability of the automated labeling process. A representative sample of 1000 CoT instances was randomly selected from the annotated dataset. Each instance was independently labeled by three large language models, including *Qwen2.5-72B-Instruct*Yang et al. (2025b), *Qwen3-32B*Yang et al. (2025a), and *ChatGPT-4o-mini*, as well as by three human annotators with relevant expertise. This multi-annotator setup enables a robust comparison between machine-generated and human-assigned cognitive labels. To evaluate alignment between different annotators, we adopt a relaxed consistency criterion inspired by educational assessment frameworks: a cognitive or knowledge label is considered acceptable if it differs from the reference label by at most one adjacent level in Bloom's taxonomy hierarchy. Prior research Thompson & Lake (2023) supports the validity of this tolerance range in cognitive categorization tasks.

To assess the confidence level of model-based labeling, we apply confidence interval estimation based on a two-step procedure.(1) We treat the majority vote among human annotators as the ground truth. For each CoT instance, we compare the label assigned by model $M$ to this reference label.

Table 5: Overview of Models by Series, Name, and Application Domain

| Series | Model Name | Application Domain |
|---|---|---|
| Qwen3 | Qwen3-0.6B | General tasks, light reasoning |
| | Qwen3-1.7B | General tasks, light reasoning |
| | Qwen3-4B | General tasks, reasoning and generation |
| | Qwen3-8B | General tasks, reasoning and generation |
| | Qwen3-14B | Advanced reasoning, complex tasks |
| | Qwen3-32B | Advanced reasoning, complex tasks |
| Phi4 | Phi4-reasoning-plus | Advanced reasoning and logic tasks |
| | Phi4-reasoning | Reasoning and analytical tasks |
| | Phi4-minireasoning | Lightweight reasoning tasks |
| DeepSeek | DeepSeek-R1-Distill-Qwen-32B | Model distillation and knowledge retrieval |
| | DeepSeek-R1-0528-Distill-Qwen3-8B | Model distillation and knowledge retrieval |
| Qwen2.5 | Qwen2.5-32B-Instruct | Instruction-tuned, general tasks |
| | Qwen2.5-14B-Instruct | Instruction-tuned, general tasks |
| | Qwen2.5-72B-Instruct | Instruction-tuned, general tasks |
| | Qwen2.5-Coder-32B-Instruct | Programming instructions, code generation |

A prediction is marked as correct if it exactly matches the human-assigned label (or falls within the acceptable tolerance range, as defined above). The model's accuracy is then calculated as the proportion of correct predictions over the total number $n$ of samples $\hat{p}$. (2) We construct a confidence interval for the observed accuracy using the binomial proportion confidence interval formula:

$$\text{CI} = \hat{p} \pm Z_{\alpha/2} \cdot \sqrt{\frac{\hat{p}(1 - \hat{p})}{n}}, \tag{4}$$

where $Z_{\alpha/2}$ refers to the Z-score corresponding to the chosen confidence level. This interval quantifies the statistical reliability of the model's annotation accuracy, offering a formal measure of the model's consistency with human judgment. The results show that the annotated model *Qwen2.5-72B-Instruct* aligns closely with human annotations and reasoning, exhibiting a very high consistency interval in Table 6 and Table 7; similar results are also observed for Cohen's Kappa ($\kappa$).

## 7.2 ANALYSIS

Table 6: The Cognitive Dimension Consistency for CI($\alpha = 95\%$)

| | Expert 1 | | Expert 2 | | Expert 3 | | Average | |
|---|---|---|---|---|---|---|---|---|
| | low | high | low | high | low | high | low | high |
| **Qwen2.5-32B-Instruct** | 0.8297 | 0.8768 | 0.8497 | 0.8939 | 0.8417 | 0.8881 | 0.8403 | 0.8863 |
| **Qwen3-32B** | 0.7537 | 0.8084 | 0.7479 | 0.8030 | 0.7536 | 0.8086 | 0.7517 | 0.8067 |
| **CompassJudger-2-32B-Instruct** | 0.8125 | 0.8642 | 0.8268 | 0.8765 | 0.8161 | 0.8687 | 0.8185 | 0.8698 |
| **Qwen2.5-72B-Instruct** | **0.8604** | **0.9036** | **0.8618** | **0.9047** | **0.8701** | **0.9127** | **0.8641** | **0.9070** |
| **ChatGPT4-mini** | 0.7537 | 0.8084 | 0.7479 | 0.8030 | 0.7536 | 0.8086 | 0.7517 | 0.8067 |

Table 7: The Knowledge Dimension Consistency for CI($\alpha = 95\%$)

| | Expert 1 | | Expert 2 | | Expert 3 | | Average | |
|---|---|---|---|---|---|---|---|---|
| | low | high | low | high | low | high | low | high |
| **Qwen2.5-32B-Instruct** | 0.9644 | 0.9873 | 0.9697 | 0.9905 | 0.9570 | 0.9831 | 0.9637 | 0.9869 |
| **Qwen3-32B** | 0.9519 | 0.9774 | 0.9591 | 0.9824 | 0.9475 | 0.9743 | 0.9528 | 0.9780 |
| **CompassJudger-2-32B-Instruct** | 0.9720 | 0.9917 | 0.9685 | 0.9896 | 0.9687 | 0.9902 | 0.9697 | 0.9905 |
| **Qwen2.5-72B-Instruct** | **0.9792** | **0.9956** | **0.9792** | **0.9956** | **0.9783** | **0.9954** | **0.9789** | **0.9955** |
| **ChatGPT4-mini** | 0.9519 | 0.9774 | 0.9591 | 0.9824 | 0.9475 | 0.9743 | 0.9528 | 0.9780 |

# 8 TEST DETAILS

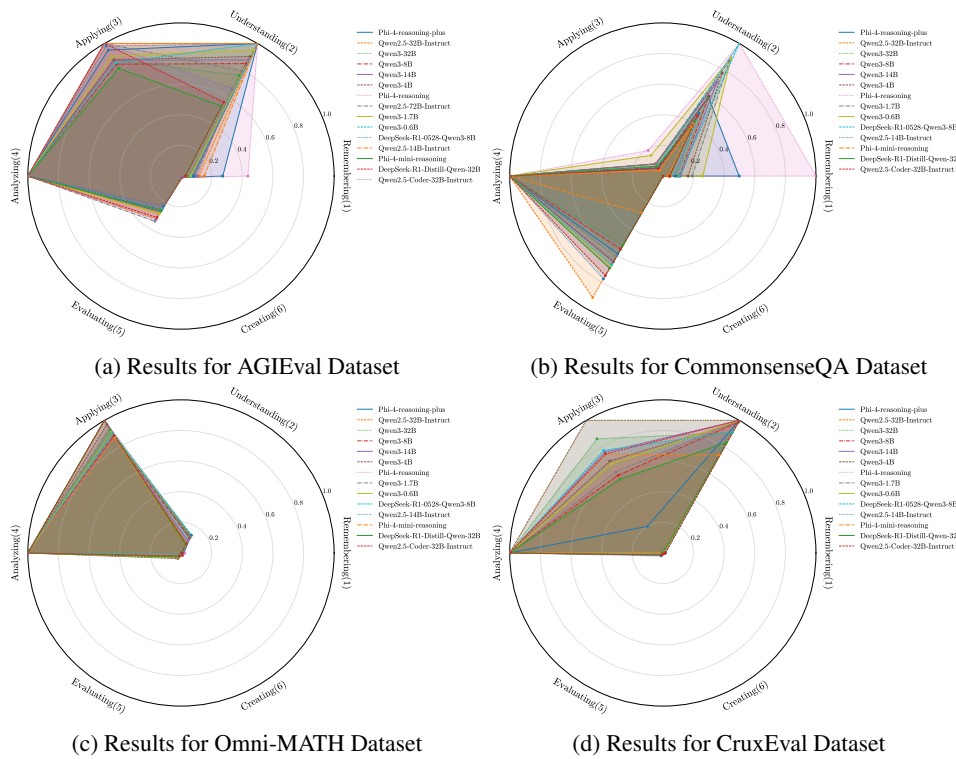

(a) Results for AGIEval Dataset   (b) Results for CommonsenseQA Dataset

(c) Results for Omni-MATH Dataset   (d) Results for CruxEval Dataset

Figure 7: The whole details for the LLM's distilled CoT results.

Figure 8 illustrates how the average cognitive level of distilled CoT reasoning in the Qwen3 series changes with model scale. Larger models generally produce reasoning with higher cognitive scores, but the improvement varies by benchmark. CommonsenseQA shows the most pronounced increase, Omni-MATH remains consistently high with marginal gains, AGIEval rises gradually, and CruxEval peaks early for smaller models before stabilizing. This indicates that scaling benefits reasoning performance, but its effect is task-dependent.

# 9 INSTRUCTION PROMPTS

## 9.1 PROMPTS TEMPLATE

We used the same system prompt as our evaluation target, and then employed different user prompts to specify various annotation tasks. The system prompt is shown as follows:

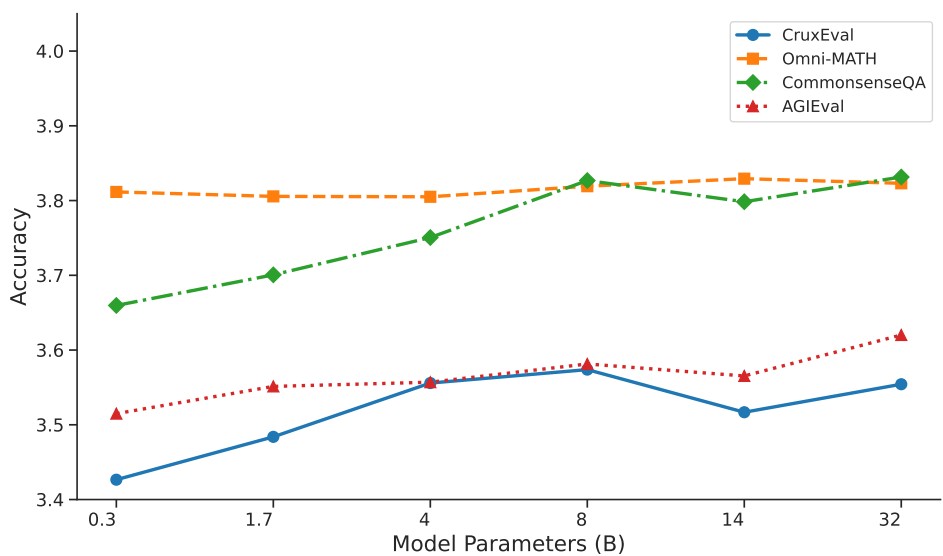

Figure 8: Average cognitive level scores of distilled CoT reasoning from the Qwen3 LLM series across different model scales on four benchmarks.

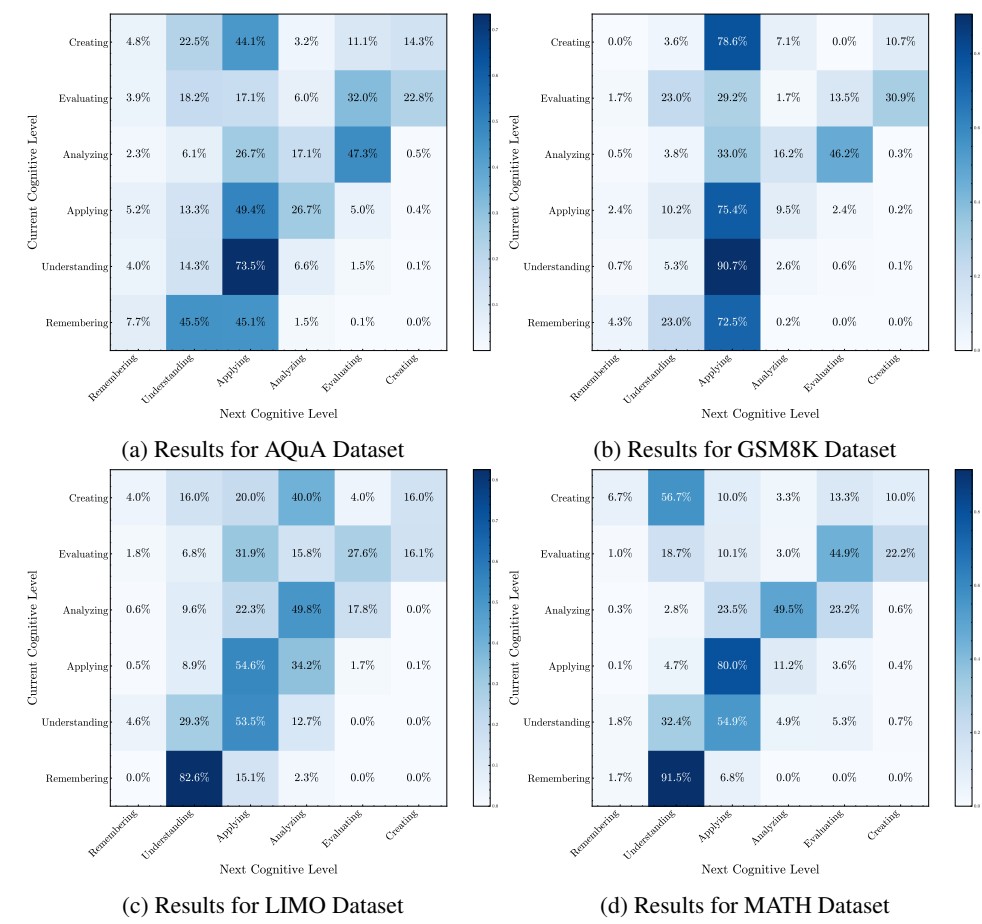

(a) Results for AQuA Dataset

(b) Results for GSM8K Dataset

(c) Results for LIMO Dataset

(d) Results for MATH Dataset

Figure 9: Cognitive trajectory annotation results: CoT reasoning mainly spans lower-order to mid-order cognition, with rare transitions to higher-order processes.

972
973
974

| **System Prompt** |
| --- |
| You are a helpful assistant facilitating meaningful dialogue between users and assistants. The user poses a question, and the assistant provides a solution by first reasoning through the problem before delivering a response. Please make sure to display the complete thought process in your outputs, including `<think></think>` in think sections, `<answer></answer>` in answer section. **Example Output:** `<think>thinking process</think><answer>Final answer</answer>` |

975
976
977
978
979
980
981
982
983
984
985
986
987
988
989
990
991
992
993
994
995
996
997
998
999
1000
1001
1002
1003
1004
1005
1006
1007
1008
1009
1010
1011
1012
1013
1014
1015
1016
1017
1018
1019
1020

**Instruction cognitive annotations** Instruction-level cognitive annotation refers to the labeling of the average cognitive processes within a chain of thought, as defined in detail in the main text. In the user prompts, we provide much more detailed definitions of Bloom's cognitive processes and the corresponding annotation methods. To ensure the annotation process is more accurate, the prompts guide the model to first reason and then provide an answer, resulting in more precise responses. The user prompt is shown as follows:

1021
1022
1023
1024
1025

| **User Prompt** |
|---|

Assume you are a data annotation expert. Please use the "question" and the "thinking process" to classify the following "thinking process" according to Bloom's Taxonomy cognitive framework. First, explain your reasoning and analytical process for the classification, then provide its corresponding cognitive process dimension and knowledge dimension.

The revised version of the Bloom's taxonomy is divided into two dimensions: cognitive and knowledge.The cognitive dimensions include:

- **Remembering**: Thinking that focuses on retrieving or recognizing previously learned information from memory, such as recalling specific facts, terms, dates, or basic concepts without necessarily interpreting them.

- **Understanding**: Thinking that involves processing and interpreting information to demonstrate comprehension, such as explaining a concept in one's own words, summarizing a text, or classifying items into categories based on their meaning.

- **Applying**: Thinking that centers on using acquired knowledge, skills, or procedures in new or practical situations, such as solving a problem with a learned method, executing a task based on a rule, or adapting a concept to a different context. It bridges theory and practice.

- **Analyzing**: Thinking that entails breaking down complex information into its individual parts to examine relationships, patterns, or underlying structures. This could include comparing ideas, identifying causes and effects, or organizing data to reveal insights about how the pieces fit together.

- **Evaluating**: Thinking that involves assessing or critiquing information, arguments, or methods based on specific standards or criteria. Examples include judging the reliability of a source, weighing the strengths and weaknesses of an approach, or determining the quality of a solution.

- **Creating**: Thinking that focuses on combining or synthesizing elements to produce something new and original, such as designing a project, constructing a novel solution, or generating innovative ideas by integrating prior knowledge in unique ways.

The dimensions of knowledge include:

- **Factual**: Basic elements such as terminology, facts, and discrete pieces of information (the "what").

- **Conceptual**: Relationships among ideas, theories, models, and structures (the "why").

- **Procedural**: How to do something—methods, techniques, and criteria for using skills and algorithms (the "how").

- **Metacognitive**: Awareness and regulation of one's own cognition—strategies for learning and self-assessment (the "knowing about knowing").

Please **categorize the following think process with bloom's taxonomy**,**DONOT solve THE PROBLEM**, provide your thought process,and then give the answer. Here is an example:

**User Prompt**

```
========================================
Input:
```json
{
"question": "<Question>",
"think\_process":"<Thinking>",
"answer":"<answer>"
}
```
Output: <think> Why fits Bloom's Taxonomy some levels
</think><answer>("cognitive","knowledge")</answer>
========================================
```

Write the answers in the tags with format (cognitive, knowledge), and there is only one tag for cognitive dimension and one tag for knowledge dimension,DO NOT GENERATE OTHER IR-RELATIVE THINGS, and multiple tags cannot be generated. Now begin your inputs:

**Trajectory cognitive annotations** The **cognitive trajectory annotation** captures the cognitive labels of individual reasoning steps within a *CoT*. We defined detailed Bloom's cognitive definitions and then divided them into several parts according to the chain-of-thought process, mapping each part to its corresponding cognitive level. The user prompt is shown as follows:

| **System Prompt** |
|---|

You will be given a long chain-of-thought (CoT) reasoning text. Your task is to segment this text into a series of clear, logically complete reasoning steps, and annotate each step with its corresponding level in Bloom's taxonomy. The cognitive dimensions include:

- **Remembering**: Thinking that focuses on retrieving or recognizing previously learned information from memory, such as recalling specific facts, terms, dates, or basic concepts without necessarily interpreting them.

- **Understanding**: Thinking that involves processing and interpreting information to demonstrate comprehension, such as explaining a concept in one's own words, summarizing a text, or classifying items into categories based on their meaning.

- **Applying**: Thinking that centers on using acquired knowledge, skills, or procedures in new or practical situations, such as solving a problem with a learned method, executing a task based on a rule, or adapting a concept to a different context. It bridges theory and practice.

- **Analyzing**: Thinking that entails breaking down complex information into its individual parts to examine relationships, patterns, or underlying structures. This could include comparing ideas, identifying causes and effects, or organizing data to reveal insights about how the pieces fit together.

- **Evaluating**: Thinking that involves assessing or critiquing information, arguments, or methods based on specific standards or criteria. Examples include judging the reliability of a source, weighing the strengths and weaknesses of an approach, or determining the quality of a solution.

- **Creating**: Thinking that focuses on combining or synthesizing elements to produce something new and original, such as designing a project, constructing a novel solution, or generating innovative ideas by integrating prior knowledge in unique ways.

Do not omit any part of the original content.Each step should represent a distinct unit of thought, such as a single observation, inference, recall, or comparison.

You are only allowed to segment the original CoT content into multiple parts and assign a single Bloom's taxonomy label to each part.

The output must be **a sequence containing only individual Bloom-level labels**. **WARNING: DONOT GENERATE ANY OTHER IRRELEVANT CONTENTS!** **The output can only be a python list and cannot contain any other irrelevant content! And any steps should be one of Bloom's taxonomy tags.**

===============================================

Here is an Example:

**System Prompt**

Input:

```json
{
"question": "<Question>",
"think\_process":"<Thinking>",
"answer":"<answer>"
}
```

Output: `<think>` why the annotation tags is set, give the thinking for annotation `</think>`

`<answer>`

```
[
["bloom-cognitiveA","raw_text"],
["bloom-cognitiveB","raw_text"],
...,
["bloom-cognitiveN","raw_text"]
]
```

`</answer>`

============================================

Now begin the normal Input:

---

**Algorithm 1:** Cognitive Annotation of Chain-of-Thought

---

**Input:** Dataset $D = \{x_1, ..., x_i, ..., x_N\}$
**Output:** Average cognitive annotation $D_I$, Cognitive trajectory annotation $D_T$
**Initialization:**
Set average cognitive annotation prompt $p_I$ and cognitive trajectory annotation prompt $p_T$;
Load dataset $D$ and format into $x_I = (q, CoT, a, p_I)$, $x_T = (q, CoT, a, p_T)$;
Initialize annotated sets $D_I = \emptyset$, $D_T = \emptyset$;
Initialize remaining sets $D_E^{(I)} = D_E^{(T)} = D$;
**for** $j \in \{I, T\}$ **do**
    **while** $D_E^{(j)} \neq \emptyset$ **do**
        Sample a batch$(i : i + B)$ from $D_E^{(j)}$: $x_{i:i+B} = (q_{i:i+B}, CoT_{i:i+B}, a_{i:i+B}, p_{i:i+B}^{(j)})$;
        $O_{i:i+B} \leftarrow LLM(x_{i:i+B}^{(j)})$;
        **for** $k \in (i : i + B)$ **do**
            **if** $O_k$ *satisfies annotation requirements* **then**
                $D^{(j)} \leftarrow D^{(j)} \cup O_{x_k^{(j)}}$;
                $D_E^{(j)} \leftarrow D_E^{(j)} - O_{x_k^{(j)}}$;

**return** $D_I, D_T$

---

## 9.2 CASE STUDY

Here are some examples of annotated thought chains:

---

**A case for Instruction-Level Annotation**

"question": "Harly's animal shelter has 80 dogs. She adopts out 40% of them but then has to take back 5 because of personality conflicts with other dogs in their adopted homes. How many dogs does she have now?", "think_process": "How many dogs are adopted out? ** First find how many dogs are adopted out: `80 dogs * 40% = <<80*40*.01=32>>32 dogs` How many dogs does Harly have now? ** Then subtract the number of dogs adopted out from the total number of dogs and add the number who were returned to find the final number of dogs: `80 dogs - 32 dogs + 5 dogs = <<80-32+5=53>>53 dogs`", "answer":"53", "response": "`<think>`

Why this fits Bloom's Taxonomy:

+ The person calculates the number of dogs adopted out by applying the percentage to the total number of dogs (this shows they can use mathematical procedures).

+ They then adjust the total number of dogs by subtracting those adopted out and adding those returned (this shows they can apply operations to solve a problem).

+ They are using procedural knowledge — methods, techniques, and criteria for using skills and algorithms to solve a practical problem.

`</think><answer>(Applying,Procedural)</answer>`"

---

**A case for Instruction-Level Annotation**

"question": "Meso can type 15 pages in 5 minutes. Tyler can type the same 15 pages in 3 minutes. How many minutes would it take Meso and Tyler to type 40 pages working together?",

"think_process": "Meso types 15 pages / 5 minutes = `<<15/5=3>>3` page per minute.

Tyler types 15 pages / 3 minutes = `<<15/3=5>>5` pages per minute.

Together they can type 3 + 5 = `<<3+5=8>>8` pages per minute.

Working together, it would take them 40 pages / 8 pages per minute = `<<40/8=5>>5` minutes",

"answer":"5", "response": "`<think>`

Why this fits Bloom's Taxonomy:

+ The person calculates the typing speed of Meso and Tyler by dividing the number of pages by the time it takes (this shows they can apply mathematical operations to solve a problem).

+ They then combine the typing speeds to find the total pages per minute when working together (this shows they can apply the concept of combining rates).

+ Finally, they calculate the time it would take to type 40 pages together by dividing the total pages by the combined rate (this shows they can apply the concept of time, rate, and work relationships).

`</think><answer>(Applying,Procedural)</answer>`"

**A case for Trajectory-Level Annotation**

"question": "Michael is traveling on the interstate at an average speed of 50km/hr (taking into account all the necessary stops for gas, relaxation, etc.). If he goes on like this for 3 days, and the distance between Alaska and Texas is 6000 km, what percentage of this distance has he covered?", "answer": "There are 24 hours in each day, so he has been traveling for a total of `3*24 = <<24*3=72>>72` hours

At a rate of 50km/hr he has covered `50*72 = <<50*72=3600>>3600 km`

3600km as a percentage of 6000km is `(3600/6000)*100% = 60%`",

"answer":"60" `<think>` Step 1: Recall basic time conversion (hours in a day) and multiplication to find total travel time. Step 2: Apply the formula distance = speed × time to calculate total distance traveled. Step 3: Apply percentage calculation formula to find what fraction of the total distance has been covered. `</think>` `<answer>` [ ["Remembering","There are 24 hours in each day, so he has been traveling for a total of 3*24 = 72 hours."], ["Applying","At a rate of 50km/hr he has covered 50*72 = 3600 km."], ["Applying","3600km as a percentage of 6000km is (3600/6000)*100] `</answer>`

**A case for Trajectory-Level Annotation**

"question": "To make 1 liter of juice, Sam needs 5 kilograms of oranges. Each kilogram of oranges costs $3. How much money would Sam have to spend to make 4 liters of juice?", "think_process": "How many kilograms of oranges does Sam need to make 4 liters of juice? ** To make 4 liters of juice, Sam needs `4 * 5 = <<4*5=20>>20` kilograms of oranges. How much would 20 kilograms of oranges cost? ** If one kilogram of oranges costs $3, then 20 kilograms would cost `20 * 3 = $<<20*3=60>>60`.", "answer":"60", "response": "`<think>` The CoT reasoning involves a series of calculations where Sam determines the total amount of oranges needed and the total cost. Each calculation step applies a learned method (multiplication) to a practical scenario, which aligns with the **Applying** level in Bloom's taxonomy. No steps involve merely recalling facts, interpreting concepts, analyzing structures, evaluating, or creating something new; they are all straightforward applications of arithmetic to solve a problem. Therefore, every step is labeled as **Applying**. `</think>` `<answer>` [ ["Applying","How many kilograms of oranges does Sam need to make 4 liters of juice?"], ["Applying","To make 4 liters of juice, Sam needs 4 * 5 = 20 kilograms of oranges."], ["Applying","How much would 20 kilograms of oranges cost?"], ["Applying","If one kilogram of oranges costs $3, then 20 kilograms would cost 20 * 3 = $60."] ] `</answer>`"

