# OpenReview forum: "Higher-Order Cognitive Chain-of-Thought Is Enough: Evaluation, Analysis, and Optimization"
_ICLR.cc/2026/Conference — Submitted to ICLR 2026_

### Official Review · Reviewer_WDuK · 2025-10-27

**Soundness:** 2
**Presentation:** 2
**Contribution:** 1
**Rating:** 2
**Confidence:** 4

**Summary:**

This paper aims to evaluate the quality of Chain-of-Thought (CoT) data from the perspective of cognitive theory. The authors employ Bloom's Taxonomy as the core theoretical framework, categorizing the reasoning processes within CoT data along two dimensions: "cognitive process" and "knowledge type." Based on this framework, the paper conducts an empirical analysis of existing CoT datasets, identifying a distributional bias towards intermediate-order cognitive operations (e.g., "Applying" and "Analyzing"). Furthermore, the authors propose a "cognitive-guided CoT data enhancement approach" and demonstrate through experiments that supplementing with even a small amount of high-order cognitive CoT data can significantly improve model performance.

**Strengths:**

1. The attempt to apply a classic cognitive science theory (Bloom's Taxonomy) to evaluate the CoT data quality of large language models is an innovative research angle. Compared to traditional evaluation methods that focus merely on accuracy or the length of the reasoning chain, this paper seeks to explore the cognitive depth of the reasoning process itself.

2. The paper provides a quantitative analysis of the cognitive distribution across multiple existing CoT datasets, revealing a potential problem: a widespread lack of higher-order cognitive capabilities (e.g., "Evaluating" and "Creating"). This finding is insightful for the future construction of high-quality reasoning datasets.

**Weaknesses:**

1. Limitations of the Theoretical Framework: Relying on Bloom's Taxonomy as the sole theoretical foundation may oversimplify the multifaceted nature of cognitive processes in complex reasoning. Bloom's Taxonomy is primarily designed for classifying educational objectives. It is questionable whether it can sufficiently and accurately capture the complex cognitive activities manifested in LLM-generated CoT (e.g., analogical reasoning, counterfactual thinking, or optimization under multiple constraints).
2. Reliability of the Evaluation Methodology: The core methodology of this paper relies on using an LLM (Qwen2.5-72B-Instruct) as an annotator (LLM-as-Judge) to perform the cognitive classification of CoT. Although Table 1 provides a consistency check (Cohen's Kappa), this cannot fully substitute for rigorous methodological validation. Given the inherent subjectivity and complexity of classifying cognitive levels, the methodological validity and reliability remain a major weakness of this study.
3. Issues with Framework Granularity/Discrimination: The empirical results show that most CoT datasets are concentrated at similar cognitive levels ("Applying" and "Analyzing"). This raises a critical question: If datasets known to perform differently on downstream tasks appear highly similar under this framework, does it imply that the classification framework lacks sufficient granularity to distinguish between reasoning processes of varying quality?
4. Limited Contribution of the "Data Enhancement Approach": The proposed "cognitive-guided data enhancement approach" appears to be more of an empirical observation (i.e., that a small amount of high-order CoT data improves performance) rather than a systematic, reproducible method. The paper fails to provide a concrete, operational procedure to guide researchers on how to generate, identify, or select this "high-order cognitive CoT data." Therefore, designating it as an "approach" is insufficient.
5. Potential Confounding Variables: The experiment (Figure 6) suggests that high-order CoT data leads to performance gains. However, this phenomenon is likely confounded by other variables. High-order cognitive tasks often imply higher problem difficulty, longer contexts, or more complex logical structures. The study fails to adequately disentangle these factors, making it impossible to clearly attribute the performance improvement solely to the "cognitive level" itself.
6. Presentation and Clarity: The manuscript's presentation requires improvement. For instance, while the six cognitive process dimensions and four knowledge type dimensions from Bloom's Taxonomy are introduced, they are not sufficiently operationalized or illustrated with clear examples within the text. This makes it difficult for readers to understand their specific application in the CoT evaluation. Furthermore, there is unnecessary repetition in the abstract and the experiment section ( e.g., line 23-27 of the abstract; repeating the RQs in Section 4).

**Questions:**

1. In Observation 5, the authors state that "distilled CoT data achieves alignment with human cognitive patterns." Could the authors please provide a precise operational definition of "human cognitive patterns" in this context? Furthermore, could they elaborate on how Figure 3 is interpreted to support the conclusion of "Human-Aligned Coupling"?

2. Observation 2 states that "Existing CoT datasets exhibit systematic mismatches between cognitive and knowledge dimensions that contradict human learning patterns," whereas Observation 6 notes that "All datasets demonstrate cognitive transitions that adhere to human cognitive hierarchies." There appears to be a tension between these two observations. Could the authors clarify this potential contradiction? If the fundamental cognitive-knowledge pairings are mismatched (Obs. 2), how can the cognitive transitions simultaneously adhere to human cognitive hierarchies (Obs. 6)?

3. Regarding the methodological reliability of the LLM-as-Judge, can the authors provide further validation beyond the Kappa scores? For example, a detailed qualitative error analysis of the LLM's annotations, a comparison with established annotation protocols in cognitive science, or citations of relevant literature to support the LLM's validity for this specific cognitive classification task.

---

### Official Review · Reviewer_Damv · 2025-10-29

**Soundness:** 3
**Presentation:** 3
**Contribution:** 2
**Rating:** 6
**Confidence:** 2

**Summary:**

This paper addressed the problem: "Current Chain-of-Thought (CoT) data quality assessment methods neglect the quality of underlying reasoning processes, leading to ineffective LLM reasoning capabilities," and as a solution, proposed a CoT data quality assessment framework grounded in Bloom's Taxonomy, evaluating both cognitive process dimensions (Remembering, Understanding, Applying, Analyzing, Evaluating, Creating) and knowledge dimensions (Factual, Conceptual, Procedural, Metacognitive).

**Strengths:**

The paper proposes that the integration of Bloom's Taxonomy to analyze CoT data quality from a cognitive perspective is a highly novel and insightful approach. It moves beyond superficial metrics (like length or accuracy) to delve into the underlying reasoning processes. This cognitive grounding provides a much-needed theoretical foundation for CoT evaluation. Also, it develops a thorough evaluation framework, including both instruction-level and trajectory-level cognitive annotations. The use of LLMs (Qwen2.5-72B-Instruct) for annotation, validated by human experts with high Cohen's Kappa scores ($\kappa > 0.88$), suggests a robust and scalable method. The detailed analysis revealing systematic distributional biases in both human-authored and LLM-distilled CoT datasets (concentration in intermediate cognitive processes and misalignment of cognitive-knowledge dimensions) is a really good contribution. This diagnosis is crucial for understanding current LLM reasoning limitations. The proposed cognitive-guided data enhancement approach, demonstrating that a small, high-quality subset of higher-order CoT data can significantly improve LLM performance, offers a practical and efficient strategy for future CoT dataset construction. This challenges the "bigger is better" paradigm.

**Weaknesses:**

- While the paper demonstrates the effectiveness of adding "minimal higher-order cognitive CoT data," the specifics of how this data is constructed or selected for optimal impact could be elaborated further. For example, are there specific guidelines or heuristics for generating these crucial ~300 examples?
- While Bloom's Taxonomy is a well-established framework, its direct applicability and nuances when strictly mapping complex, emergent LLM reasoning processes might warrant further discussion, especially for highly abstract or novel tasks not explicitly covered by the taxonomy's original scope.

**Questions:**

- Could the authors provide more specific details on the selection process for the "minimal higher-order cognitive CoT data" (approximately 300 examples) that led to significant performance boosts? Are there specific criteria or a semi-automated process used?
- Given the observed cognitive-knowledge dimensional misalignment (Observation 2), how might future CoT dataset construction actively mitigate this issue beyond simply adding higher-order CoT, perhaps through guided annotation paradigms?
- The paper focuses on Bloom's Taxonomy. Have the authors considered or could they briefly discuss how their framework might interact with or be extended by other cognitive theories (e.g., those focusing on problem-solving strategies, metacognitive control, or executive functions) to provide an even richer understanding of LLM reasoning?

---

### Official Review · Reviewer_vcUf · 2025-10-31

**Soundness:** 2
**Presentation:** 2
**Contribution:** 2
**Rating:** 4
**Confidence:** 3

**Summary:**

This paper proposes a cognitive perspective-based Chain-of-Thought (CoT) data quality assessment framework to address the insufficient evaluation of reasoning process quality in current CoT datasets. The study reveals systematic biases in the distribution of cognitive operations in existing CoT data and introduces a simple yet effective cognitive-guided CoT data enhancement method that significantly improves model reasoning capabilities using minimal higher-order cognitive CoT data.

**Strengths:**

Proposed a CoT quality assessment framework based on Bloom's Taxonomy, conducting extensive experiments using various benchmark datasets and models.

**Weaknesses:**

The experiments mainly focus on mathematical reasoning tasks(Table 2,3,4), lacking coverage in other domains. Mathematical reasoning is a very distinct part of cognitive theory.

Although the paper emphasizes human cognition, it lacks direct comparison with human performance (only annotations from three experts were used).

The analysis of differences in cognitive characteristics across different task domains is rather brief.

Some figures have large proportions but lack substantial information (e.g., Figure 1 and Figure 3).

**Questions:**

Why not use professional datasets from the field of cognitive science (and real human experiments)?

The paper states that existing datasets fail to adequately represent human-level cognitive capabilities, but it seems unclear how this work significantly surpasses prior works in this aspect.

**Details Of Ethics Concerns:**

Lack of annotator compensation

---

### Meta-Review · Area_Chair_2eoS · 2026-01-05

**Summary:**

This paper proposes a cognitive-theory-grounded framework for assessing Chain-of-Thought (CoT) data quality using Bloom’s Taxonomy, and reports an empirical analysis showing that existing CoT datasets are biased toward mid-level cognitive operations. It further claims that adding a small amount of higher-order (e.g., evaluative or metacognitive) CoT data can improve downstream model performance. Reviewers generally agreed that the perspective is interesting and somewhat novel, but they diverged substantially on whether the framework is sufficiently rigorous and whether the contributions go beyond high-level observations. The scores are mixed (2/4/6), with one reviewer leaning weak accept (Damv), one marginal reject (vcUf), and one clear reject (WDuK), reflecting concerns about methodological validity, theoretical fit, and clarity. Given the lack of author response or rebuttal, I lean toward rejection.

**Reviewer Concerns:**

Since the authors did not respond during the rebuttal or discussion phase, none of the substantive concerns raised by reviewers were addressed. Key outstanding issues include limited domain coverage beyond math reasoning (vcUf), lack of comparison to actual human cognitive performance and reliance on a small number of annotators (vcUf), and insufficient operational detail on how higher-order CoT data is constructed or selected (Damv, WDuK). More critically, Reviewer WDuK raised concerns about the suitability and granularity of Bloom’s Taxonomy for capturing complex LLM reasoning, the reliability of the LLM-as-judge annotation methodology beyond reported kappa scores, and potential confounds in attributing gains specifically to “cognitive level” rather than difficulty or length. These concerns remain unresolved and significantly weaken the paper’s claims.

**Reviewer Scores:**

Reviewer vcUf (score 4) might have increased slightly with a strong rebuttal clarifying human cognition claims and ethics/annotation details, but without responses would likely remain unchanged. Reviewer Damv (score 6) explicitly asked for more concrete methodology and theoretical discussion; absent any clarification, I expect their score would likely drop to a borderline or neutral position. Reviewer WDuK (score 2) raised fundamental objections to the framework and methodology, and without rebuttal there is no reason to expect any upward revision. Overall, the lack of engagement prevents convergence, and the balance of concerns supports a reject recommendation.

---

### Decision · Program_Chairs · 2026-01-26

Reject